# SEMI-SUPERVISED GENERATIVE MODELING FOR CONTROLLABLE SPEECH SYNTHESIS

**Raza Habib**[1]*  **Soroosh Mariooryad**[2]  **Matt Shannon**[2]  **Eric Battenberg**[2]  **RJ Skerry-Ryan**[2]
**Daisy Stanton**[2]  **David Kao**[2]  **Tom Bagby**[2]

[1]University College London (UCL)    [2]Google Research.

raza.habib@cs.ucl.ac.uk

{soroosh, mattshannon, ebattenberg, rjryan, daisy, davidkao, tombagby}@google.com

## ABSTRACT

We present a novel generative model that combines state-of-the-art neural text-to-speech (TTS) with semi-supervised probabilistic latent variable models. By providing partial supervision to some of the latent variables, we are able to force them to take on consistent and interpretable purposes, which previously hasn't been possible with purely unsupervised TTS models. We demonstrate that our model is able to reliably discover and control important but rarely labelled attributes of speech, such as affect and speaking rate, with as little as 30 minutes supervision. Even at such low supervision levels we do not observe a degradation of synthesis quality compared to a state-of-the-art baseline. Audio samples are available on the web[1].

## 1 INTRODUCTION

The ability to reliably control high level attributes of speech, such as emotional expression (affect) or speaking rate, is often desirable in speech synthesis applications. Achieving this control however is made difficult by the necessity of acquiring a large quantity of high quality labels. In this paper we show that semi-supervised latent variable models can take us a significant step closer towards solving this problem.

Combining state-of-the-art neural text-to-speech (TTS) systems with probabilistic latent variable models provides a natural framework for discovering aspects of speech that are rarely labelled or even difficult to describe. Both inferring the latent prosody and generating samples with sufficient variety requires reasoning about uncertainty and is thus a natural fit for deep generative models.

There has been recent progress in applying stochastic gradient variational Bayes (SGVB) (Kingma & Welling, 2013; Rezende et al., 2014) to training probabilistic neural TTS models. Battenberg et al. (2019) and Hsu et al. (2018) have shown that it is possible to use latent variable models to discover features such as speaking style, speaking rate, arousal, gender and even the quality of the recording environment.

However, these models are formally non-identifiable (Hyvärinen & Pajunen, 1999; Locatello et al., 2019) and this implies that repeated training runs will not reliably discover the same latent attributes. Even if they did, a lengthy human post-processing stage is necessary to identify what the model has learned on any given training run. In order to be of practical use for control, it is not enough for the models to discover latent attributes, they need to do so reliably and in a way that is robust to random initialization and to changes in the model. We demonstrate that the addition of even modest amounts of supervision can be sufficient to achieve this reliability.

By augmenting state-of-the art neural TTS with semi-supervised deep generative models within the VAE framework (Kingma et al., 2014; Narayanaswamy et al., 2017), we show that it is possible to not only discover latent attributes of speech but to do so in a reliable and controllable manner. In particular we are able to achieve reliable control over affect, speaking rate and F0 variation (F0 is the

---

*Work performed while interning at Google Research.

[1]https://google.github.io/tacotron/publications/semisupervised_generative_modeling_for_controllable_speech_synthesis/

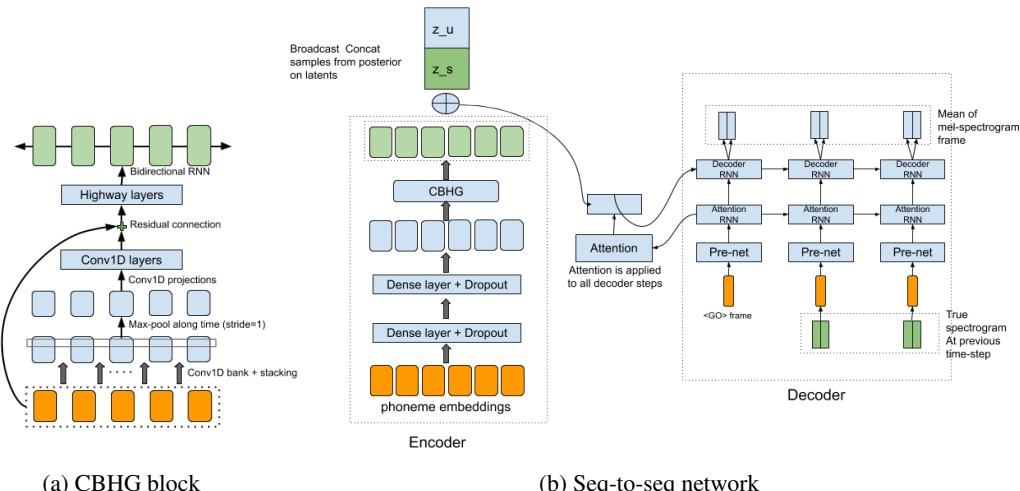

(a) CBHG block  (b) Seq-to-seq network

Figure 1: Schematic showing how we parameterize the conditional likelihood $p(x|y, z_u, z_s)$. Left: A block of 1-D convolutions and RNNs originally introduced by Wang et al. (2017) and described in detail in the appendix. Right: Schematic of the sequence-to-sequence network that outputs the means of our auto-regressive distribution. At each decoder time step, the network outputs the means for the next two spectrogram frames.

fundamental frequency of oscillation of the vocal folds). Further, we provide demonstrations that it is possible to transfer controllability to speakers for whom we have no labels. Our core contributions are:

- To combine semi-supervised latent variable models with Neural TTS systems, producing a system that can *reliably* discover attributes of speech we wish to control.

- To demonstrate that as little as 30 minutes supervision can be sufficient to improve prosody and allow control over speaking rate, fundamental frequency (F0) variation and affect, a problem of interest to the speech community for well over two decades (Schröder, 2001).

- To imbue TTS models with control over affect, F0 and speaking rate whilst still maintaining prosodic variation when sampling.

## 2 GENERATIVE MODEL

Our generative model, shown in figures 1 and 2a, consists of an autoregressive distribution over a sequence of acoustic features, $x_{1...t}$, that are generated conditioned on a sequence of text, $y_{1...k}$, and on two latent variables, $z_u$ and $z_s$. The latent variables can be discrete or continuous. $z_s$ represents the variations in prosody that we seek to control and is semi-supervised. $z_u$ is fully unobserved and represents latent variations in prosody (intonation, rhythm, stress) that we wish to model but not explicitly control. Once trained, our model can be used to synthesize acoustic features from text. Similar to Tacotron 2 (Shen et al., 2018), we then generate waveforms by training a second network such as WaveNet (van den Oord et al., 2016) or WaveRNN (Kalchbrenner et al., 2018) to act as a vocoder. In our case we use WaveRNN.

We parameterize our likelihood $p(x_{1...t}|y_{1...k}, z_u, z_s, \theta)$ by a sequence-to-sequence neural network with attention (Shen et al., 2018; Graves, 2013; Bahdanau et al., 2014) that is shown schematically in figure 1. Details largely follow Tacotron (Wang et al., 2017) and are given in appendix A. At each time step we model a mel-spectrogram frame with a fixed variance isotropic Laplace distribution whose mean is output by the neural network. We condition each of the latent variables by concatenating the vectors $z_u$ and $z_s$ to the representation of the text-encoder, before the application of the attention mechanism. In the case of continuous $z$ we use a standard normal prior and in the case of discrete $z$ we use a uniform categorical prior with one-hot encoding.

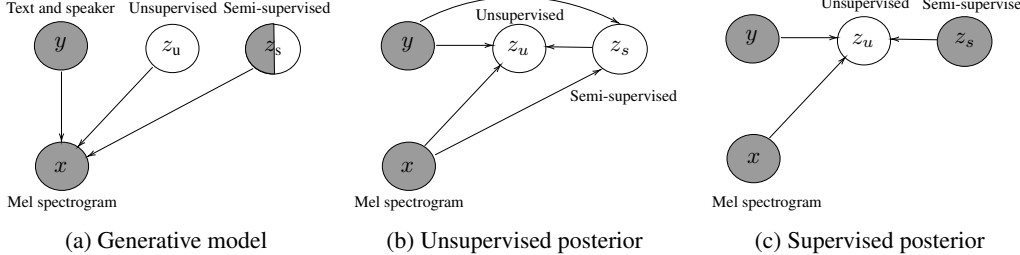

(a) Generative model   (b) Unsupervised posterior   (c) Supervised posterior

Figure 2: Left: The graphical model showing the conditional independence assumptions between each of the stochastic variables. Centre: The structure of the variational distribution used to approximate the posterior for fully unsupervised data points and Right: supervised points.

## 2.1 SEMI-SUPERVISED TRAINING

Following Kingma et al. (2014); Narayanaswamy et al. (2017), we train our model via stochastic gradient variational Bayes (SGVB). That is we approximately maximize the log-likelihood of our training data by maximizing a variational lower bound using stochastic gradient ascent. Since we are training with semi-supervision we in fact need two lower bounds: one for the data points for which $z_s$ is observed; one for the case where $z_s$ is unobserved. In our models the fully latent variable $z_u$ is always continuous but the semi-supervised latent $z_s$ can be continuous or discrete. The conditional independence structure of our variational distributions is shown in figures 2b and 2c. On supervised data, the per-datapoint bound is:

$$
\begin{aligned}
\log p(x, z_s | y, \theta) &= \log \int p(x, z_u, z_s | y, \theta) \, \mathrm{d}z_u \\
&\geq \mathbb{E}_{q(z_u | x, y, z_s, \phi)} \left[ \log \left( \frac{p(x | y, z_u, z_s, \theta) p(z_u) p(z_s)}{q(z_u | x, y, z_s, \phi)} \right) \right] \\
&= \mathbb{E}_{q(z_u | x, y, z_s, \phi)} \left[ \log p(x | y, z_u, z_s, \theta) \right] + \log p(z_s) - D_{\mathrm{KL}}(q(z_u | x, y, z_s, \phi) \| p(z_u)) \\
&= \mathcal{L}_s(\theta, \phi; x, y, z_s)
\end{aligned}
$$

Where $q(z_u | x, y, z_s, \phi)$ is a parametric variational distribution introduced to approximately marginalize $z_u$. $\theta$ are the parameters of the generative model and $\phi$ are the parameters of the variational distributions. The intractable integrals are approximated with reparameterized samples. For the cases where $z_s$ is unobserved and discrete, the bound is:

$$
\log p(x | y, \theta) = \log \int \sum_{z_s} p(x, z_u, z_s | y, \theta) \, \mathrm{d}z_u \tag{1}
$$

$$
\geq \sum_{z_s} q(z_s | x, y, \phi) \mathcal{L}_s(\theta, \phi; x, y, z_s) + H(q(z_s | x, y, \phi)) \tag{2}
$$

$$
= \mathcal{L}_u(\theta, \phi; x, y) \tag{3}
$$

and when $z_s$ is continuous we replace the sum above with an integral and again approximate with reparameterized samples. The variational distributions are parameterized by a neural network that takes as input the text, spectrograms and other conditioning variables and outputs the parameters of the distribution. The exact structure of this network is given in appendix A. We have implicitly assumed that $q(z_u, z_s | x, y, \phi)$ may be factorized as $q(z_u, z_s | x, y, \phi) = q(z_u | x, y, z_s, \phi) q(z_s | x, y, \phi)$ with shared parameters between these two distributions (see appendix A). Optimizing the variational objective with respect to the parameters $\phi$ encourages the variational distributions to match the posterior of the generative model $p(z_u, z_s | x, y, \theta)$. Unlike previous work (Hsu et al., 2018), we do not assume that the posterior on the latents is independent of the text, as this dependence likely exists in the model due to explaining away. That is to say that although the text and the latents are independent in our prior, observing the spectrogram correlates them in the posterior because they

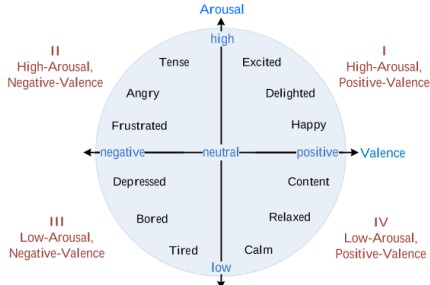

Figure 3: The circumplex model of emotion. Each possible emotion is represented in a 2 dimensional plane consisting of an arousal dimension and valence dimension. This figure is borrowed from Munoz-de Escalona & Canas (2017).

both explain variation in the spectrogram. This has been shown to be significant by Battenberg et al. (2019).

If we define:

$$\tilde{q}(z_s|x,y) = \begin{cases} q(z_s|x,y,\phi) & \text{if unsupervised} \\ \gamma\delta(z_s - z_{s_{\text{observed}}}) & \text{if supervised} \end{cases} \qquad (4)$$

then we can write the overall objective over both the supervised and unsupervised points succinctly as[2]:

$$\mathcal{L}(\theta,\phi) = \mathbb{E}_{x,y}\left[\sum_{z_s}\tilde{q}(z_s|x,y,\phi)\mathcal{L}_s(\theta,\phi;x,y,z_s) + H(\tilde{q}(z_s|x,y,\phi))\right] \qquad (5)$$

where summation would again be replaced by integration for continuous $z_s$ and $\gamma$ (shown in equation 4) is a weighting factor that pre-multiplies the loss for any supervised point. This weighting was also used in previous work such as Narayanaswamy et al. (2017), who showed it to be beneficial at very low levels of supervision.

Writing the objective in this form allows an intuitive interpretation for the semi-supervised training procedure. When supervision is provided, our objective function is evaluated at the observed value of $z_s$. When supervision is not provided, we evaluate the objective function for every possible value of $z_s$ and take a (potentially infinite for continuous $z_s$) weighted average. The weighting in the average is given by $q(z_s|x,y,\phi)$, which is simultaneously trained to approximate the posterior $p(z_s|x,y,\theta)$. In other words, on unsupervised utterances, we evaluate our objective for each possible value of the latent attribute and weight by the (approximate) posterior probability that this value of the latent was responsible for generating the utterance.

As $q(z_s|x,y,\phi)$ is trained to approximate $p(z_s|x,y,\theta)$ we can expect it to become a reasonable classifier/regressor for the semi-supervised latent attribute as the model improves. For example when $z_s$ represents an affect label, $p(z_s|x,y,\theta)$ is the posterior probability, of the model, over affect given text and speech. By taking the most likely posterior class, this distribution can be used as an affect classifier. However, this variational distribution is only trained on unsupervised training points and so does not benefit directly from the supervised data. To overcome this problem we follow Kingma et al. (2014) and add a classification loss to our objective. The overall objective becomes:

$$\mathcal{L}_{\text{total}}(\theta,\phi) = \mathcal{L}(\theta,\phi) + \alpha\mathbb{E}_{x,y,z_s}[\log q(z_s|x,y,\phi)] \qquad (6)$$

Where $\alpha$ is a hyperparameter which adjusts the contribution of this term.

---

[2]We define the differential entropy of the delta function to be 0

## 3 DATA

We have used a proprietary high quality labeled data-set of 40 English speakers. The training set consists of 72,405 utterances with durations of at most 5 seconds (45 hours). The validation and test sets each contain 745 utterances or roughly 30 minutes of data. We vary the amount of supervision in the experiments below. We also experimented with transferring controllability to a fully unlabeled data-set of audiobook recordings by Catherine Byers (the speaker from the 2013 Blizzard Challenge), which exhibits high variation in affect and prosody and to other speakers who were less expressive. We strongly encourage the reader to listen to the synthesized samples on our demo page[3].

In this work we chose to focus on learning to control affect with a discrete representation, as well as speaking rate and F0 variation with a continuous representation, as these are challenging aspects of prosody to control. Our method could be applied to other factors without modification.

### 3.1 AFFECT CONTROL

The best way to represent emotion is an actively researched area and many models of affect exist. In this work we chose to follow the circumplex model of emotion (Russell, 1980) which posits that most affective states can be represented in a 2 dimensional plane with one axis representing arousal and the other axis representing valence. Arousal measures the level of excitement or energy and valence measures positivity or negativity. Figure 3, shows a chart of emotions plotted in the arousal-valence plane where we can see that, for example, high arousal and high valence corresponds to joy or happiness whereas high arousal and low valence might correspond to anger or frustration.

Our data-set was recorded under studio conditions with trained voice actors who were prompted to read dialogues in one of three valences:-2, -1, +2 and two arousal values:-2 (low), +2 (high). This was achieved by prompting the actors to read dialogues in either a happy, sad or angry voice at two levels of arousal. This results in 6 possible affective states which we chose to model as discrete and use as our supervision labels.

### 3.2 SPEAKING RATE AND F0 VARIATION CONTROL

In order to demonstrate that we can control continuous attributes we also created approximate real-valued labels for speaking rate and arousal for all of our data. We generate the approximate speaking rate as number of syllables per second in each utterance. F0, also known as the fundamental frequency, measures the frequency of vibration of the vocal folds during voiced sounds. Variation in F0 is highly correlated with arousal and roughly measures how expressive an utterance is. To create approximate arousal labels we extracted the F0 contour from each of our utterances, using the YIN algorithm (De Cheveigné & Kawahara, 2002), and measured its standard deviation. We then performed a whitening transform on these two approximate labels in order to match it to our standard normal prior.

These artificial labels would of course be cheap to obtain for the entire data-set and would not justify the use of semi-supervision in real applications. But, our objective here is to evaluate/demonstrate the efficacy of semi-supervision rather than to specifically control a particular attribute. We have chosen syllable rate and F0 standard deviation, because they both correspond to subjectively distinct variations of interest, and they are more easily quantifiable than affect and so provide strong evidence of controllability. For the continuous latents we are not only able to interpolate speaking-rates and F0 variations but also to extrapolate outside of our training data. We provide examples on our demo page of samples with significantly greater/lower speed and F0 variation than typically observed in natural speech.

## 4 EXPERIMENTS AND RESULTS

To evaluate the efficacy of semi-supervised latent variable models for controllable TTS we trained the model described in section 2 on the above data-sets at varying levels of supervision as well as for

---

[3]Sound demos are available at `https://google.github.io/tacotron/publications/semisupervised_generative_modeling_for_controllable_speech_synthesis/`.

|  |  | baseline vs. angry | Valence
baseline vs. sad | baseline vs. happy | Arousal
low vs. high |
|---|---|---|---|---|---|
| preference
score | 27 min (1%) | $-0.20 \pm 0.10$ | $-0.60 \pm 0.08$ | $-0.43 \pm 0.09$ | $-0.50 \pm 0.09$ |
|  | 135 min (5%) | $-0.74 \pm 0.07$ | $-0.83 \pm 0.06$ | $-0.83 \pm 0.06$ | $-0.57 \pm 0.08$ |
|  | 270 min (10%) | $-0.71 \pm 0.07$ | $-0.86 \pm -0.95$ | $-0.61 \pm 0.08$ | $-0.59 \pm 0.08$ |

Table 1: Subjective metrics for affect control. Negative is a preference for the controlled model. +1 and -1 indicate a preference for samples A and B, respectively. For valence, raters are told that a sample is intended to convey a particular emotion, e.g. happy, and then presented with sample from baseline without control (A), and controlled model (B), and asked to choose between them. For arousal, raters are told to choose the sample that is more vocally aroused, and presented with controlled samples in low (A) and high (B) arousal. To avoid bias, the orders are randomly altered during rating. We show preference score and 95% confidence intervals at multiple supervision levels.

varying settings of the hyperparameters: $\alpha$ which controls the supervision loss and $\gamma$, which over emphasizes supervised training points. We found that a value of $\alpha = 1$ was optimal for the discrete experiments and $\alpha = 0$ for the continuous experiments, which corresponds to simply optimizing the ELBO. For each experiment we report the results for the best $\gamma$ found, and $\gamma = 1$. $\gamma = 1$ corresponds to experiments with no over-weighting of the supervised points. All models were trained using the ADAM optimizer with learning rate of $10^{-3}$ and run for $300,000$ training steps with a batch size of 256, distributed across 32 Google Cloud TPU chips. All models were implemented using tensorflow 1 (Abadi et al., 2016).

Assessing the degree of control is challenging as interpreting affect is subjective. We used two objective metrics of control as well as subjective evaluation from human raters and a third objective metric of overall quality. For affect, the first objective metric we introduced was the test-set accuracy of a 6-class affect classifier trained on the ground truth training data and applied to generated samples from the model (shown in figure 4a). The classifier is a convolutional neural network whose structure mirrors the posterior network $q(z_s|x,y,\phi)$ and its exact architecture is given in appendix A. We also provide subjective metrics of controllability, shown in table 1. For speaking rate control, we measure the mean syllable rate error on a held out test-set. The syllable rate error is calculated as the absolute difference in syllable rate between the desired syllable rate and that measured from the synthesized sample. We calculate an analogous error rate for F0 variation.

Whilst the two metrics above measure controllability they don't tell us if this comes at the expense of a degradation in synthesis quality. To probe quality we use three further metrics. The first was Mel-Cepstral-Distortion-Dynamic-Time-Warping (MCD-DTW) (Kubichek, 1993) on a held out test-set, shown in figure 4d. MCD-DTW is a measure of the difference between the ground-truth spectrogram and the synthesized mel spectrogram that is known to correlate well with human perception (Kubichek, 1993). The second metric of quality was crowd sourced mean-opinion-scores (MOS). The third metric of quality is speech recognition word error rate (WER) and character error rate (CER) on audio samples. The MOS and speech recognition results are summarized in table 2.

To demonstrate that semi-supervision by including unlabelled data is beneficial, we also provide MOS and speech recognition errors for fully supervised subsets of the data in table 2. These show that at least close to 5 hours of data is required to train a reasonable quality TTS model, far above the 30 minutes supervision needed to control prosodic aspects of speech.

We provide further details of all of these metrics in the appendix B, and sample spectrograms are provided in appendix C.

## 5 DISCUSSION

The classification accuracy (see figure 4a), subjective metrics (see table 1) and error-rate results (see figure 4b-4c) provide a clear demonstration that using semi-supervised latent variables, we are able to achieve control of both continuous and discrete attributes of speech. There is not a significant degradation in the overall quality and this is evidenced by the mean opinion scores which are above the baseline, Tacotron, and also speech recognition errors (see table 2). We also include a baseline of our Tacotron model augmented only by the unsupervised latent $z_s$, to aid comparison.

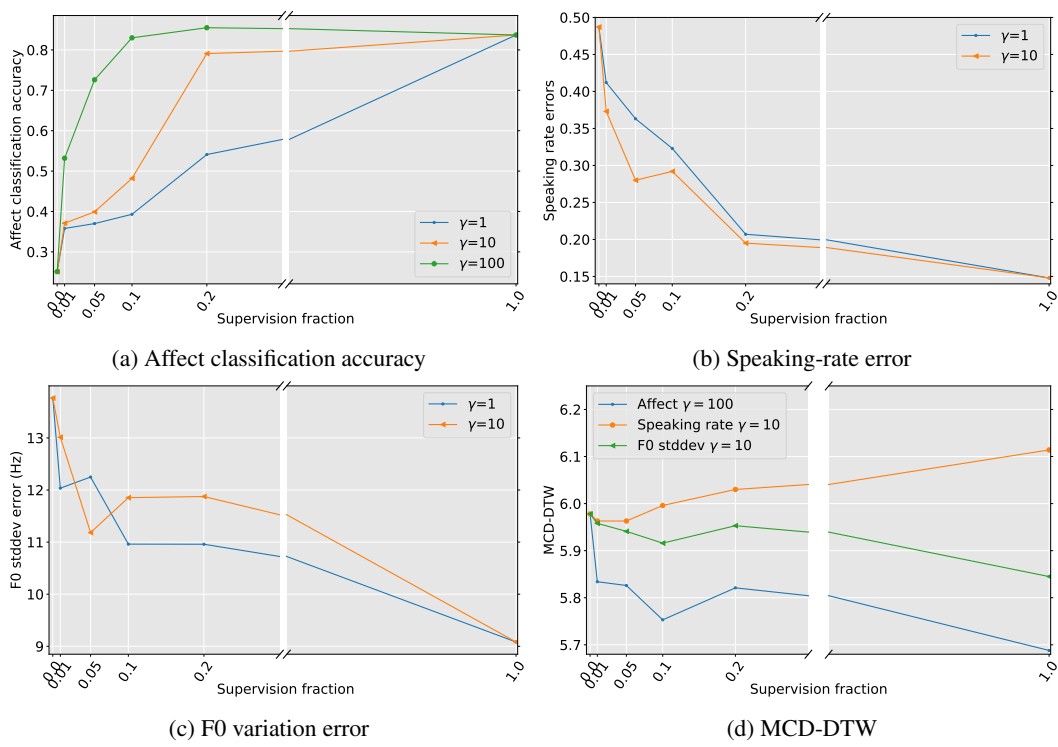

(a) Affect classification accuracy

(b) Speaking-rate error

(c) F0 variation error

(d) MCD-DTW

Figure 4: Objective evaluation metrics as a function of supervision fraction. $100\%$ supervision corresponds to 45 hours of supervised training data and $0\%$ supervision corresponds to base tacotron. For MCD-DTW and error-rates lower is better.

| | | | | Semi-Supervised ($10\%$ supervision) | | |
| | | | | continuous latent | | discrete latent |
| | ground truth | baseline | baseline with $z_u$ | F0 | speaking-rate | affect |
|---|---|---|---|---|---|---|
| MOS | $4.52\pm0.07$ | $4.09\pm0.09$ | $4.24\pm0.08$ | $4.28\pm0.07$ | $4.16\pm0.08$ | $4.17\pm0.09$ |
| WER | 4.49 | 4.93 | 4.93 | 4.06 | 2.53 | 6.09 |
| CER | 2.11 | 2.38 | 2.22 | 1.80 | 1.14 | 3.12 |

Table 2: Metrics of overall quality: Mean Opinion Scores (MOS) alongside 95% confidence intervals, speech recognition word error rate (WER), and chararacter error rate (CER). The results show no degradation in performance compared to the baseline.

| | 27 min (1%) | 54 min (2%) | 108 min (4%) | 135 min (5%) | 270 min (10%) | 45 hours (100%) |
|---|---|---|---|---|---|---|
| MOS | unintelligible | unintelligible | $3.20\pm0.13$ | $3.52\pm0.11$ | $4.03\pm0.08$ | $4.08\pm0.09$ |
| WER | 91.95 | 96.31 | 19.83 | 7.55 | 5.56 | 4.93 |
| CER | 74.57 | 78.9 | 12.54 | 4.46 | 2.79 | 2.22 |

Table 3: Metrics of overall quality for fully supervised data at varying data-set sizes, showing significant degradation below 270 minutes.

The MCD-DTW scores for F0 variation and affect are improved at all levels of supervision (figure 4d). Whilst the MCD-DTW is degraded for speaking rate, this is likely a misleading metric when targeting changes in timing as the dynamic-time-warping component of MCD-DTW changes exactly the aspect we wish to control. For speaking rate the combination of MOS and samples is a better indication of the overall quality. We are able to reduce the supervision level to levels as low as $1\%$ or 30 minutes and still have a significant degree of control. We show on our demo page[4] that even

---

[4] https://google.github.io/tacotron/publications/semisupervised_generative_modeling_for_controllable_speech_synthesis/

at 3 minutes of supervision we can still achieve control of speaking rate and that we are able to extrapolate outside the range of values seen during training. On the affect data our classification accuracy doesn't degrade significantly until we reach $10\%$ (300 minutes) supervision and remains significantly above chance down to levels as low as $1\%$ (30 minutes), see figure 4a and table 1. Obtaining 30 minutes of supervised data is likely within reach of most teams constructing TTS systems. Unlike previous work on generative modelling for control (Hsu et al., 2018; Wang et al., 2018), we do not require a post-processing stage to determine what our latent variables control and we can pre-determine what aspects we wish to control through choice of data. By separating our latent variables into those that are partially supervised and those which are fully unsupervised we retain the ability to model other latent aspects of prosody; this means that we can still draw samples of varying prosody whilst holding constant the affect, speaking rate or F0 variation.

We observe the greatest degree of affect control, as measured by classifier accuracy, when $\alpha = 1$ and $\gamma = 100$. This means that to achieve the highest controllability we needed to 1) provide extra information to our approximate posterior $q(z_s|x, y, \phi)$ and 2) to over-represent the supervised data at low levels of supervision. Although both of these hyperparameters have been used in the literature before (Narayanaswamy et al., 2017; Kingma et al., 2014) and shown to be either beneficial or necessary, they aren't strictly required by our probabilistic framework and so it is worth considering why they are needed. There are three potential sources of error in any generative model trained with SGVB: the model itself may be mis-specified such that the true data-generating distribution is not in the model class, the parametric family chosen to approximate the posterior may be overly restrictive and finally the optimization landscape may contain undesirable local minima. These problems have afflicted previous work with deep latent variable models trained with SGVB, resulting in models that don't use their latent variables unless trained with complex annealing schedules (Bowman et al., 2015). In our case we believe that the necessity to set $\alpha$ and $\gamma$ arises from a combination of model mis-specification and local minima. If $\alpha$ is set to 0, then at the start of training $q(z_s|x, y, \phi)$ is trained only to approximate $p(z_s|x, y, \theta_0)$, which is randomly initialized. We found empirically that in our discrete-latent experiments this resulted in $q(z_s|x, y, \phi)$ collapsing early in training to a point-mass on a single class. Having ended up in this undesirable local minimum the posterior distribution never recovered, despite this being an obviously poor approximation to the model posterior later in training. The addition of the classification loss and supervision weighting were sufficient to overcome this collapse and allow $q$ to continue to model the posterior.

The optimization landscape is strongly affected by the relative size of the conditional likelihood and KL terms in our objective. These are in turn strongly affected by our choice of conditional independence assumptions and output-distributions. Thus, a natural direction for further work is to increase the expressivity of the conditional likelihood $p(x|y, z_s, z_u, \phi)$ to reduce model mis-specification. This could be done by learning the variance of the Laplace-distribution we currently use or by parameterizing more expressive output distributions that do not assume conditional independence across spectrogram channels. We conjecture that with more expressive output distributions, it may be possible to reduce the need for the $\alpha$ and $\gamma$ terms in the objective. In this work we chose to use quite simple unconditional diagonal Gaussian priors, as our primary goal was to demonstrate the practicality of semi-supervision. Another natural extension would be to use conditional-priors $p(z|y)$ and to use more expressive priors such as mixtures as was done in Hsu et al. (2018).

## 5.1 RELATED WORK

There has been enormous recent progress in neural TTS with numerous novel models proposed in recent years to synthesize speech directly from characters or phonemes (Shen et al., 2018; Arik et al., 2017; Gibiansky et al., 2017; Ping et al., 2017; Vasquez & Lewis, 2019; Taigman et al., 2017). Differentiating factors between these models include the degree of parallelism, with some models using Transformer based architectures (Ren et al., 2019), the choice of conditional independence assumptions made (Vasquez & Lewis, 2019) or the number of separately trained components (Gibiansky et al., 2017). Our work here is largely orthogonal to the exact structure of the conditional likelihood $p(x|y, z_s, z_u)$ and could be combined with all of the above methods.

Much of the recent research focus has been on modeling latent aspects of prosody. Early attempts include Global Style Tokens (Wang et al., 2018) which attempted to learn a trainable set of style-embeddings. Wang et al. (2018) condition the Tacotron decoder on a linear combination of embedding vectors whose weights during training are predicted from the ground-truth spectrogram.

They were able to achieve prosodic control but there is no straightforward way to sample utterances of varying prosody. More recently, attempts have also been made to combine probabilistic latent variable models trained using SGVB (Akuzawa et al., 2018; Wan et al., 2019). These models use a fully unsupervised and non-identifiable approach, which makes it difficult to disentangle or interpret their latent variables for control. Hsu et al. (2018) attempt to overcome this problem by using a Gaussian mixture as the latent prior and so perform clustering in the latent space. Battenberg et al. (2019) introduce a hierarchical latent variable model to separate the modelling of style from prosody. However, all of these methods are fully unsupervised and this results in latents that can be hard to interpret or require complex post-processing.

The work most similar to ours is Wu et al. (2019) which also attempts to achieve affect control using semi-supervision with a heuristic approach based on Global Style Tokens (Wang et al., 2018). Wu et al. (2019) add a cross-entropy objective to the weightings of the style-tokens that encourages them to be one-hot on points with supervision. Similar to our method, they are able to achieve control over affect but unlike our method they do not have a principled probabilistic interpretation nor the ability to simultaneously model aspects of prosody other than emotion. The result is that their method is not able to draw samples of varying prosody for the same utterance with fixed emotion. Furthermore, whilst our method can be applied to both continuous and discrete controllable factors, its not clear how to extend the style-token based approach to handle continuous latent factors.

In the wider generative modelling literature, the combination of semi-supervision and deep latent variable models was first introduced in Kingma et al. (2014) who focus on using unlabelled data to improve classification accuracy. The potential to use the same technique for controllable generation was recognized by Narayanaswamy et al. (2017) who also provided demonstrations on image synthesis tasks. Since that work, interest in learning disentangled latent variables has grown but generally pursued alternate directions such as re-weighting the ELBO (Higgins et al., 2017), augmenting the objective to encourage factorization (Kim & Mnih, 2018) or using adversarial training (Mathieu et al., 2016). The ability to transfer controllability to speakers for whom we do not have supervision is referred to as domain transfer and our model bears similarities to that introduced by Ilse et al. (2019) but they use a mixture in their latent space more similar to Hsu et al. (2018).

## 5.2 ETHICAL CONSIDERATIONS

As with many advances in speech synthesis, progress in controllability raises the prospect that bad actors may misuse the technology either for misinformation or to commit fraud. Improvements in data efficiency and realism increase these risks and, when publishing, a consideration has to be made as to whether the benefits of the developments outweigh the risks. It is the opinion of the authors in this case that, since the focus of this work is on improved prosody, with potential benefits to human-computer interfaces, the benefits likely outweigh the risks. We nonetheless urge the research community to take seriously the potential for misuse both of this work and broader advances in TTS.

## 6 CONCLUSION

We have shown that the combination of semi-supervised latent variable models with neural TTS presents a practical and principled path towards building speech synthesizers we can control. Unlike previous fully unsupervised methods, we are able to consistently and reliably learn to control predetermined aspects of prosody. Our method can be applied to any latent attribute of speech for which a modest amount of labelling can be obtained, whether it be continuous or discrete. In our experiments we found that 30 minutes of supervision was sufficient, a volume of data that is within the reach of most research teams. We are able to learn to control subtle characteristics of speech such as affect and for continuous attributes we have provided demonstrations of extrapolation to ranges never seen during training, and to speakers with no supervision. Augmenting existing state-of-the-art TTS systems with latent variables does not degrade synthesis quality and we evidence this with crowd sourced mean opinion scores. Unlike similar heuristic methods, our probabilistic formulation, allows us to draw samples of varying prosody whilst holding constant some attribute we wish to control.

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

# A  NEURAL NETWORK ARCHITECTURE

| Module | hyperparameters |
|---|---|
| Input | Text normalized phonemes |
| Phoneme embedding | 256-D |
| Pre-net | FC-256-Relu-Dropout(0.5) |
| | → FC-128-Relu-Dropout(0.5) |
| CBHG text encoder | Conv1D bank: K=16, conv-k-128-Relu |
| | →Max pooling with stride=1 width=2 |
| | → Conv1D projections: conv-3-128-Relu |
| | → conv-3-128-Linear |
| | → Highway net: 4 layers of FC-128-Relu |
| | → Bidirectional GRU: 128 cells |
| Attention type | 5 component GMM attention w/ softplus (Graves, 2013) |
| Attention RNN | LSTM-256-Zoneout(0.1) → FC-128-tanh |
| DecoderRNN | 2-layer residual-LSTM-265-zoneout(0.1) |
| | → FC-80-Linear |
| Frames-per-timestep (reduction factor) | 2 |
| WaveRNN | 5 layers DilatedConv1D-512 |
| | → 2 layers TransposeConv + ReLu |
| | → GRU-768 conditioned on 5 previous samples |
| | → FC-768-relu |
| | → 3 component MoL, 24kHz sample rate |
| Variational posterior | Spectrogram |
| | → 6 Conv-layers 32-32-64-64-128-128 |
| | → LSTM-128 |
| | → FC-128-tanh |
| Optimizer | ADAM with learning rate $10^{-3}$, batch-size 256 |
| Speaker embedding | 64-D |

Table 4: Summary of the hyperparameters described below.

**Sequence-to-sequence model** Our sequence-to-sequence network is modelled on Tacotron (Wang et al., 2018) but uses some modifications introduced in Skerry-Ryan et al. (2018). Input to the model consists of sequences of phonemes produced by a text normalization pipeline rather than character inputs. The CBHG text encoder from Wang et al. (2017) is used to convert the input phonemes into a sequence of text embeddings. The phoneme inputs are converted to learned 256-dimensional embeddings and passed through a pre-net composed of two fully connected ReLU layers (with 256 and 128 units, respectively), with dropout of 0.5 applied to the output of each layer, before being fed to the encoder. For multi-speaker models, a learned embedding for the target speaker is broadcast-concatenated to the output of the text encoder. The attention module uses a single LSTM layer with 256 units and zoneout of 0.1 followed by an MLP with 128 tanh hidden units to compute parameters for the monotonic 5-component GMM attention window. We use GMMv2b attention mechanism described in Battenberg et al. (2020). Instead of using the exponential function to compute the shift and scale parameters of the GMM components as in Graves (2013), GMMv2vb uses the softplus function, and also adds initial bias to these parameters, which we found leads to faster alignment and more stable optimization. The attention weights predicted by the attention network are used to compute a weighted sum of output of the text encoder, producing a context vector. The context vector is concatenated with the output of the attention LSTM layer before being passed to the first decoder LSTM layer. The autoregressive decoder module consists of 2 LSTM layers each with 256 units, zoneout of 0.1, and residual connections between the layers. The spectrogram output is produced using a linear layer on top of the 2 LSTM layers, and we use a reduction factor of 2, meaning we predict two spectrogram frames for each decoder step. The decoder is fed the last frame of its most recent prediction (or the previous ground truth frame during training) and the current context as computed by the attention module. Before being fed to the decoder, the previous prediction is passed through a pre-net with the same same structure used before the text encoder above but its own parameters.

**CBHG text encoder**   We reuse the CHGB text encoder introduced in Wang et al. (2018). The text encoder consists of a bank of 1-D convolutional filters, followed by highway networks and a bidirectional gated recurrent unit (GRU) recurrent neural net (RNN). The input sequence is first convolved with K sets of 1-D convolutional filters, where the $k$-th set contains $C_k$ filters of width $k$. The convolution outputs are stacked together and further max pooled, preserving time. As in the original paper we use a stride of 1 to preserve the original time resolution. We further pass the processed sequence to a few fixed-width 1-D convolutions, whose outputs are added with the original input sequence via residual connections. Batch normalization is used for all convolutional layers. The convolution outputs are fed into a multi-layer highway network to extract high-level features. Finally, we stack a bidirectional GRU RNN on top to extract sequential features from both forward and backward context.

**Variational posteriors**   The variational distributions $q(z_s|x, y)$ and $q(z_u|x, y, z_s)$ are both structured as diagonal Gaussian distributions whose mean and variance are parameterized by neural networks. For discrete supervision we replace $q(z_s|x, y)$ by a categorical distribution and use the same network to output just the mean. The input to the distribution starts from the mel spectrogram $x$ and passes it through a stack of 6 convolutional layers, each using ReLU non-linearities, 3x3 filters, 2x2 stride, and batch normalization. The 6 layers have 32, 32, 64, 64, 128, and 128 filters, respectively. The output of this convolution stack is fed into a unidirectional LSTM with 128 units. We pass the final output of this LSTM (and potentially vectors describing the text and/or speaker) through an MLP with 128 tanh hidden units to produce the parameters of the diagonal Gaussian posterior which we sample from. All but the last linear layer of these networks is shared between the two distributions $q(z_s|x, y)$ and $q(z_u|x, y, z_s)$ . The resulting sample is broadcast-concatenated to the output of the text encoder. In our experiments $z_u$ is always 32-dimensional and $z_s$ is either a one-hot vector across 6 classes or a 1 dimensional continuous value.

**Conditional inputs**   When providing information about the text to the variational posterior, we pass the sequence of text embeddings produced by the text encoder to a unidirectional RNN with 128 units and use its final output as a fixed-length text summary that is passed to the posterior MLP. Speaker information is passed to the posterior MLP via a learned speaker embedding.

**WaveRNN**   We used a WaveRNN model similar to that described in Kalchbrenner et al. (2018) as our vocoder. Our WaveRNN uses discretized mixture of logistics output as described in Salimans et al. (2017) instead of the dual softmax from that paper, and conditions on 5 previous samples at each step instead of only 1 previous. We trained the network to map from synthesized mel-spectrograms to waveforms, training on 900 sample windows. A conditioning stack of dilated convolution and transpose convolutions is applied to the input spectrogram before tiling to upsample to the audio sample rate.

## B   EVALUATION

**Mel spectrograms**   The mel spectrograms the model predicts are computed from 24 kHz audio using a frame size of 50 ms, a hop size of 12.5 ms, an FFT size of 2048, and a Hann window. From the FFT energies, we compute 80 mel bins distributed between 80 Hz and 12 kHz.

**MCD-DTW**   To compute mel cepstral distortion (MCD) (Kubichek, 1993), we use the same mel spectrogram parameters described above and take the discrete-cosine-transform to compute the first 13 MFCCs (not including the 0th coefficient). The MCD between two frames is the Euclidean distance between their MFCC vectors. Then we use the dynamic time warping (DTW) algorithm (Velichko & Zagoruyko, 1970) (with a warp penalty of 1.0) to find an alignment between two spectrograms that produces the minimum MCD cost (including the total warp penalty). We report the average per-frame MCD-DTW.

**Affect classifier**   The affect classifier has a very similar structure to the variational posterior. The input to the classifier starts from the mel spectrogram $x$ and passes it through a stack of 6 convolutional layers, each using ReLU non-linearities, 3x3 filters, 2x2 stride, and batch normalization. The 6 layers have 32, 32, 64, 64, 128, and 128 filters, respectively. The output of this convolution stack is

fed into a unidirectional LSTM with 128 units. The final output of the LSTM is then passed through a softmax non-linearity to get logits over the training classes. We use the same data splits described in section 3 to train and evaluate the classifier. The classifier is tuned on the validation set achieving 84.33% classification accuracy, generalizing well to the test set with 83.94% accuracy.

**Mean opinion scores**   We use a human rating service similar to Amazon's Mechanical Turk, with a large pool of English speakers to collect MOS evaluations. The MOS template is shown in figure 5. A human rater is presented with a single speech sample and is asked to rate perceived naturalness on a scale of 1 to 5, where 1 is "Bad" and 5 is "Excellent". We have selected the utterances of one male, and one female speaker in our test set, totalling 371 utterances to evaluate. For each sample, we collect 1 rating, and no rater is used for more than 6 items in a single evaluation set. In total, 270 unique raters completed the 6 evaluation sets presented in table 2. Since raters are randomly selected for each set, some raters have assessed multiple methods. Across the 6 evaluation sets, the average and median of total number of ratings per rater was 8.24, and 6, respectively. To analyze the data from these subjective tests, we average the scores and compute 95% confidence intervals. Natural human speech is typically rated around 4.5. Samples used for MOS from our model were drawn using the mean of $z_u$, whilst sampling $z_s$.

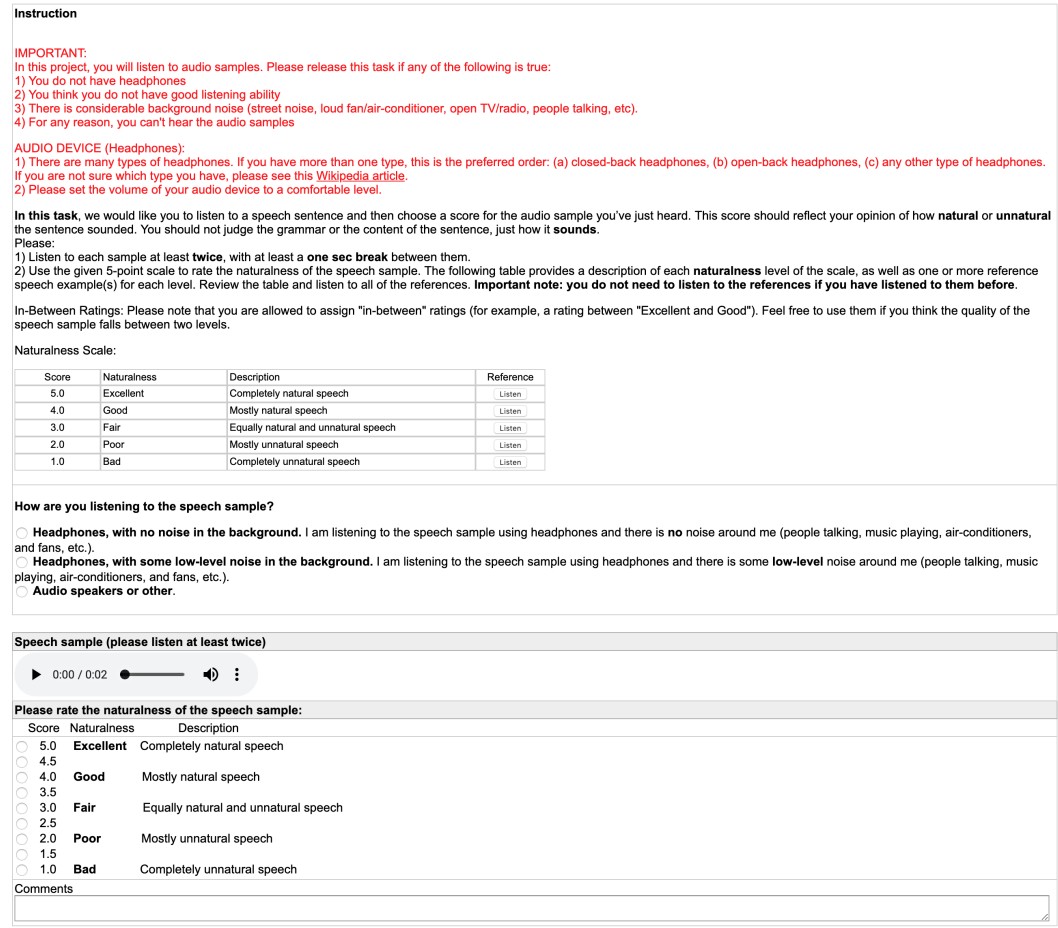

Figure 5: Mean opinion score (MOS) evaluation template. For each utterance, the human raters assign a 1–5 score of the perceived naturalness, with 1 being "Bad" and 5 being "Excellent".

**Subjective affect control evaluation**   We use the same rater pool, and the same set of 371 utterances used for the MOS evaluations. The A/B template is shown in figure 6. For each utterance, the human rater is presented with a pair of utterances to choose the one that better conveys the target emotion (e.g., happy in the figure). Both utterances are generated with the same text. To evaluate the

control over valence, we present baseline (i.e, no control) against utterances generated in specific valence category (angry, happy and sad). To evaluate the control over arousal, we present samples generated at low arousal against samples generated at high arousal, and ask the rater to choose the utterance that is more vocally aroused. We use mean of $z_u$ to generate all the samples.

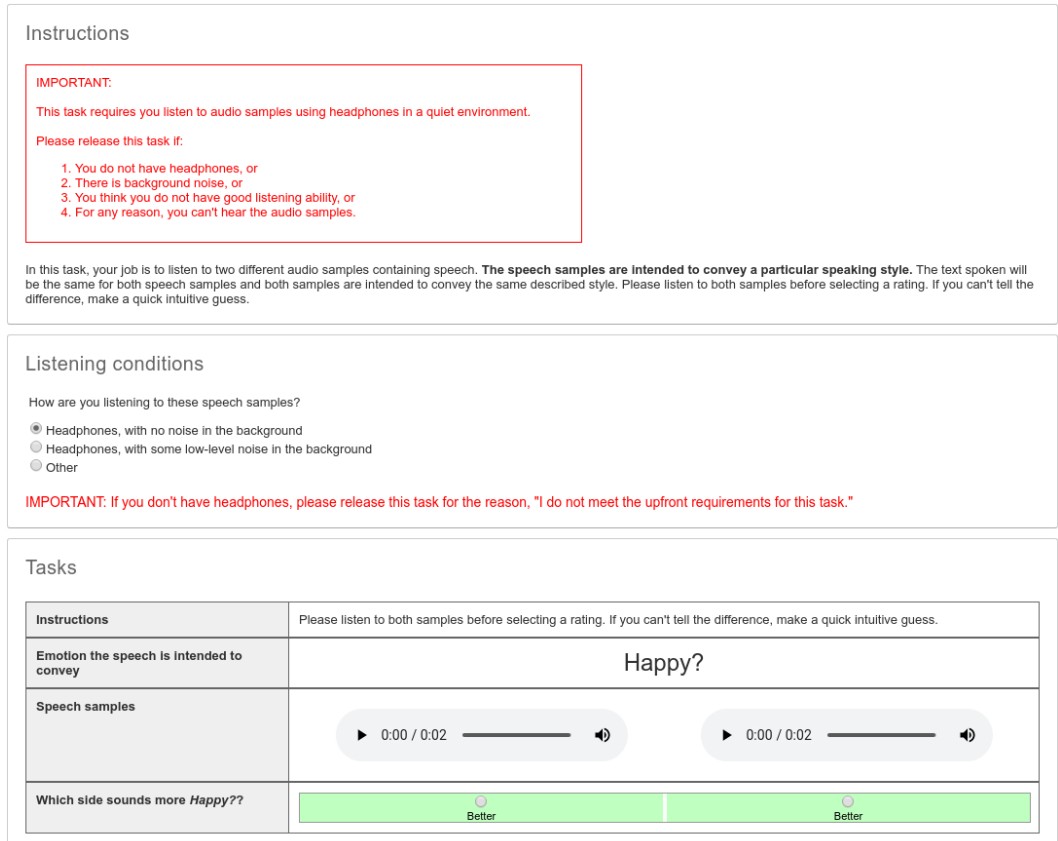

Figure 6: A/B evaluation affect control evaluation template. The emotion label (Happy in the figure) varies depending on the task.

## C  SAMPLE SPECTROGRAMS

**Controlling affect**    Table 5 shows the effect of varying the valence and arousal latent variable on the spectrogram and F0 track. We can see that a low valence for sadness corresponds to the flattest F0 track, and high arousal manifests in higher F0 values and variations.

**Controlling speaking rate and pitch variations**    Table 6 shows the effect of varying speaking rate, and F0 variation control variables on a sample spectrogram and F0 track. When controlling speaking rate (first column), the duration reduces as we increase the input speaking rate control, while the F0 variation remains stable. When controlling the F0 variation (second column), the pitch dynamic range increases, while the duration remains constant, which demonstrates controllablity and also some degree of disentanglement.

## D  REPRODUCING RESULTS ON LIBRITTS PUBLIC DATASET

To verify the reproduciblity of our results on a public dataset, we trained models to control speaking rate and F0 variation on clean subset of LibriTTS dataset (Zen et al., 2019). We only use the utterances below 5 seconds, which is 62 hours of data. We have done no tuning on this dataset, and directly used the hyperparameters we used for our internal dataset. Figures 7a and 7b show the

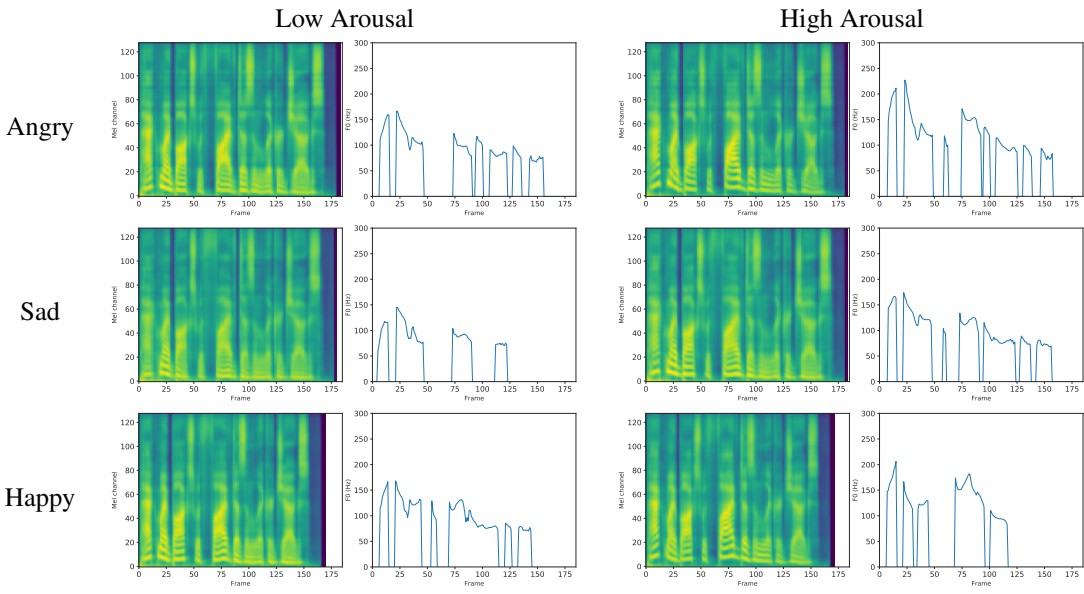

Table 5: Sample spectrogram and F0 track plots, generated by varying affect labels, valence in y-axis, and arousal in x-axis.

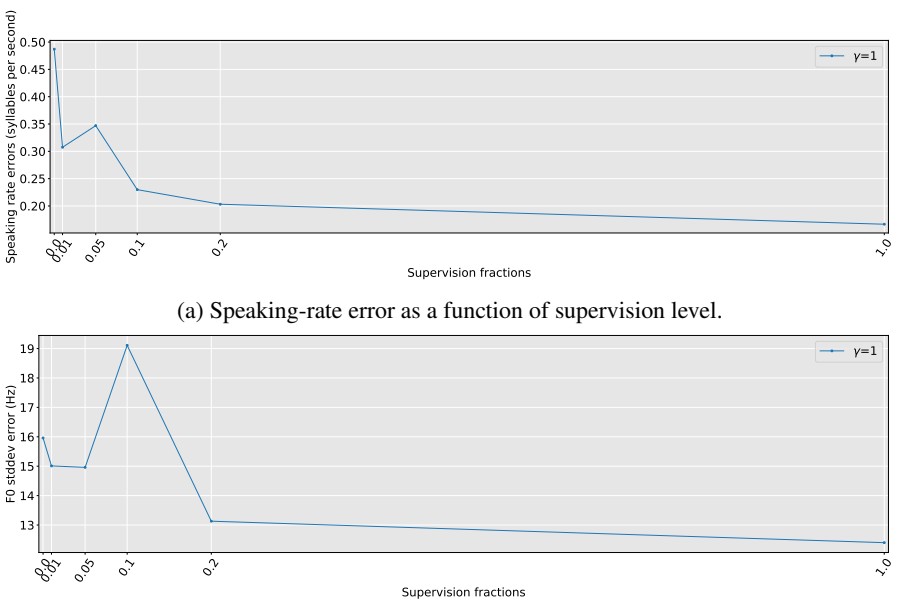

(a) Speaking-rate error as a function of supervision level.

(b) F0 variation error as a function of supervision level

Figure 7: Objective controllability of speaking rate and F0 variation evaluation metrics presented at multiple supervision levels, on LibriTTS (Zen et al., 2019) datasets. $100\%$ supervision corresponds to 62 hours of supervised data.

errors of producing the desired speaking rate, and F0 standard deviation, which generally go down as function of supervision level, with exception of $10\%$ supervision for controlling F0 variation [5]. Given, this is a lower quality dataset, with many more speakers and with much smaller data per speaker and also the fact that we have done zero hyperparameter tuning on this dataset, this result look very encouraging.

---

[5]We believe the result at $10\%$ to be anomalous and likely due to a single bad training run.

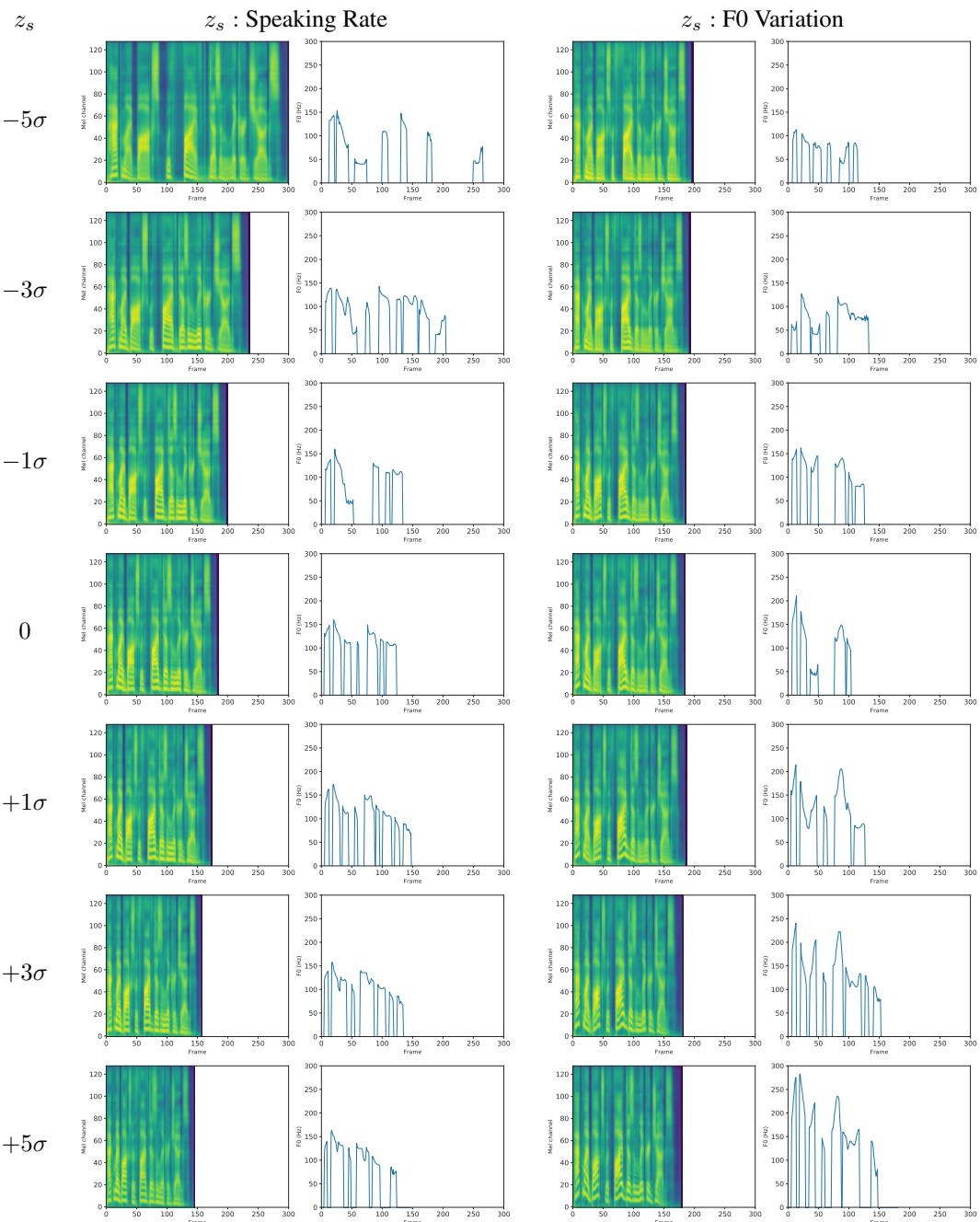

Table 6: Sample spectrogram and F0 track plots, generated by varying the speaking rate (first column) and F0 variation (second column). We use standard normal prior for these factors and this table demos varying the control factor from $-5\sigma$ to $5\sigma$, demonstrating the controllability, interpolation and extrapolation of conditional generation, and also disentanglement of these factors.

