# OpenReview forum: "Semi-Supervised Generative Modeling for Controllable Speech Synthesis"
_ICLR.cc/2020/Conference — Accept (Poster)_

### Official Review · AnonReviewer4 · 2019-10-21
**Official Blind Review #4**

**Rating:** 6

**Review:**

Overview:

This paper proposes to use semi-supervised learning to enforce interpretability on latent variables corresponding to properties like affect and speaking rate for text-to-speech synthesis. During training, only a few training items are annotated for these types of properties; for items where these labels are not given, the variables are marginalised out. TTS experiments are performed and the approach is evaluated objectively by training classifiers on top of the synthesised speech and subjectively in terms of mean opinion score.

I should note that, although I am a speech researchers, I am not a TTS expert, and my review can be weighed accordingly.

Strengths:

The proposed approach is interesting. I think it differs from standard semi-supervised training in that at test time we aren't explicitly interested in predicting labels from the semi-supervised labelled classes; rather, we feed in these labels as input to affect the generated model output. I agree that this is a principled way to impart interpretability on latent spaces which are obtained through unsupervised modelling aiming to disentangle properties like affect and speaking rate.

Weaknesses:

This work misses some essential baselines, specifically a baseline that only makes use of the (small number of) labelled instances. In the experiments, the best performance is achieved when gamma is set very high, which (I think) correspond to the purely supervised case (I might be wrong). Nevertheless, I think a model that uses only the small amount of labelled data (i.e. without semi-supervised learning incorporating unlabelled data) should also be considered.

As a minor weakness, the evaluation seems lacking in that human evaluations are only performed on the audio quality, not any of the target properties that are being changed. For affect specifically, it would be helpful to know whether the changes can be perceived by humans. As a second minor weakness, some aspects of the paper's presentation can be improved (see below).

Overall assessment:

The paper currently does not contain some very relevant baselines, and I therefore assign a "weak reject".

Questions, suggestions, typos, grammar and style:

- p. 1: "control high level attributes *of of* speech"
- p. 2: It would be more helpful to state the absolute amount of labelled data (since 1% is somewhat meaningless).
- p. 2: I am not a TTS expert, but I believe the last of your contributions have already been achieved in other work.
- Figure 2: It would be helpful if these figures are vectorised.
- p. 4: "*where* summation would again ..."
- Figure 4: Is there a reason for the gamma=1000 experiment, which performs best in (a), not to be included in (b) to (d)?
- Section 5: Table 1 is not references in the text.
- Section 5.1: "P(x|y,z_s,z_u)" -> "p(x|y,z_s,z_u)"
- In a number of places, I think the paper meant to cite [1] but instead cited the older Kingma & Welling (2013) paper; for instance before equation (6) (this additional loss did not appear in the original VAE paper).

References:

[1] https://arxiv.org/abs/1406.5298

Edit: Based on the author's response, I am changing my rating from a 'weak reject' to a 'weak accept'.

**Experience Assessment:**

I have published one or two papers in this area.

**Review Assessment: Checking Correctness Of Derivations And Theory:**

I assessed the sensibility of the derivations and theory.

**Review Assessment: Checking Correctness Of Experiments:**

I assessed the sensibility of the experiments.

**Review Assessment: Thoroughness In Paper Reading:**

I read the paper at least twice and used my best judgement in assessing the paper.

---

> ### Author Response · Authors · 2019-11-14
> **Summary of Response to Reviewer # 4**
>
> We thank the reviewer for their feedback.
>
> To address the reviewers concerns about baselines and subjective metrics we have added to the paper:
>
> 1) Word-error and MOS results for fully supervised models. Shown in the table below and added to the paper.
> 2) We have added subjective metrics of controllability in addition to classification accuracy. These show that humans are able to perceive the affect control. (see table 1 in the paper for details)
>
>  Size of training set    |    27 min (1%)   |   54 min (2%)   |  108 min (4%)  | 135 min (5%)  |  270 min (10%)  | 45 hours (100%)
> ------------------------------------------------------------------------------------------------------------------------------------------------------------
>   MOS │ unintelligible | unintelligible | 3.20 +- 0.13 |  3.52 +- 0.11 | 4.03 +- 0.08 | 4.08 +- 0.09
>   Word Error Rate %     │ 91.95  | 96.31 | 19.83 | 7.55 | 5.56 | 4.93
>   Character Error Rate % │ 74.57 | 78.9 | 12.54 | 4.46 | 2.79 | 2.22
>
> We hope that this addresses the reviewer's main concern and they may reconsider their score because of this.
>
> Our experiments show that, the minimum time required to train an intelligible multi-speaker TTS system is certainly greater than 54 minutes. After 270 minutes, MOS improves only slowly with more training data. Since we are able to achieve control with 3 minutes of labels for continuous attributes and 30 minutes for discrete attributes, this strongly motivates our semi-supervised approach in the context of controllability.

---

> > ### Author Response · Authors · 2019-11-14
> > **Detailed Response to Reviewer #4 -- part 1 of 1**
> >
> > [BASELINES]
> > As you have pointed out, semi-supervision for conditional generation, is different from typical semi-supervision for label predicting tasks. Training only on supervised samples might not be a viable solution here, since, unlike label prediction tasks, learning to generate the observation manifold usually requires more data than predicting labels does.
> >
> > You’re correct that we didn’t have a baseline trained on a small fully labelled dataset. The primary reason for not including this was that at the very lowest levels of supervision that we consider (below 30 minutes), it’s implausible to us that one could train a TTS system that functioned at all. Indeed Chung et al [1] have shown that less than 36 minutes of studio quality data was insufficient to train an intelligible system (see https://google.github.io/tacotron/publications/semisupervised/). At higher supervision levels (300 minutes say) it is possible to train a TTS system that produces intelligible speech, but not necessarily a high quality one. To verify this we have added MOS scores, and the results of running speech recognizer on samples of this model at varying amounts of data. Unsurprisingly, below 54 minutes, the model does not produce intelligible speech as shown by speech recognition results, and verified by listening to samples. Also, at 108 minutes the MOS scores are significantly lower and far below the quality one would typically hope for a production TTS system. Based on speech recognition and MOS results ~5 hours is the bare minimum that we need to train a decent TTS model. Similar results are well documented in [1]. In general the amount of supervised data we use is below the threshold typically required to train a high-quality voice at all.
> >
> > [1] Chung, Y. A., Wang, Y., Hsu, W. N., Zhang, Y., & Skerry-Ryan, R. J. (2019, May). Semi-supervised training for improving data efficiency in end-to-end speech synthesis. In ICASSP 2019-2019 IEEE International Conference on Acoustics, Speech and Signal Processing (ICASSP) (pp. 6940-6944). IEEE.
> >
> > From a practical point of view, the take away message is although the TTS model itself requires a relatively large amount of data to train, finer grain control is achievable with small amounts of supervision, something that we show here with a principled generative model.
> >
> > [Gamma = 100]
> > The high setting of gamma (100) is not equivalent to a fully supervised model but does over-emphasize the contribution of the supervised points to the loss significantly.
> >
> > >>As a minor weakness, the evaluation seems lacking in that human evaluations are only performed on the audio quality, not any of the target properties that are being changed. For affect specifically, it would be helpful to know whether the changes can be perceived by humans. As a second minor weakness, some aspects of the paper's presentation can be improved (see below).
> >
> > we hoped that the combination of objective metrics and the samples provided on our demo page (tts-demos.github.io) would be sufficient to assure the reviewer that the differences are definitely perceivable by humans. We have now also added subjective human evaluation (see table 1).
> >
> > >> p. 2: It would be more helpful to state the absolute amount of labelled data (since 1% is somewhat meaningless).
> >
> > 1% corresponds to ~30 minutes of labelled speech data or roughly 270 labelled utterances.
> > We do state both in the abstract and the discussion that 1% corresponds to ~30 minutes of training data. To further clarify, however, we’ve added it to the description of the experimental results as well.
> >
> > >>p. 2: I am not a TTS expert, but I believe the last of your contributions have already been achieved in other work.
> >
> > Although past works have been able to sample all prosodic variations (including emotion etc.), to the best of our knowledge, it hasn’t been possible to randomly sample prosodic variation in all but a small subset of attributes that are held constant (e.g., emotions). If the reviewer could please suggest a reference, we would be happy to amend the paper accordingly.
> >
> > >> Figure 4: Is there a reason for the gamma=1000 experiment, which performs best in (a), not to be included in (b) to (d)?
> > For each experiment we simply reported the fully unweighted ELBO (gamma=1) and the best results achieved with differing values of gamma. In the first experiment gamma=100  performed better than gamma=10 so we included it. In b and d it performed significantly worse and so we left it off. We would like to point out that large values of gamma may cause overfitting, which is consistent with findings of [1]. We found that the optimal value varies depending on z_s being discrete or continuous.
> >
> > [2] Narayanaswamy, S., Paige, T. B., Van de Meent, J. W., Desmaison, A., Goodman, N., Kohli, P., ... & Torr, P. (2017). Learning disentangled representations with semi-supervised deep generative models. In Advances in Neural Information Processing Systems (pp. 5925-5935).

---

### Official Review · AnonReviewer1 · 2019-10-22
**Official Blind Review #1**

**Rating:** 8

**Review:**

This paper proposes to use a semi-supervised VAE based text-to-speech (TTS) for expressive speech synthesis. The main contribution of this paper is that it can provide more explicit interpretation for the latent variable with the help of the supervised learning component. The formulations and implementations in the main body are quite high-level and we may not easily understand the technical/implementation details only with the main body but, in other words, the paper is well written to convey their main messages and of course some details are described in the appendix. The experiments show the effectiveness in terms of subjective (MOS) and objective (cepstral distance etc.) with a lot of audio examples on the demo page.

My concern for this paper is a lack of reproducibility. The paper uses the in-house data to perform their experiments and the code does not seem to be publically available. Also, the paper misses several detailed information (e.g., detailed configurations of the Wave RNN vocoder, what kind of neural network toolkits and libraries). The high computational cost ("distributed across 32 Google Cloud TPU chips") would also make the reproducibility difficult. I also would like to see whether this method can have some experimental comparisons with (Hsu et al., 2018) with their postprocessing to show the distinction in terms of the performance in addition to the functional difference.

Comments:
- In general, the font size in the figures is too small
- Figure 1: it's better to have an explanation of "CBHG". People outside the end-to-end TTS community cannot understand it.
- Can you also control the noise level as shown in (Hsu et al., 2018) but more explicitly within this framework? Controlling the noise level is quite important for end-to-end TTS, and I think this method can fit this direction because we can easily obtain the noise attribute (supervision) by data simulation or annotate the noise.
- Section 2, second paragraph y_{1...t} --> y_{1...k} (?)
- equation (6), classification loss: I think this part requires more clarifications in this timing, e.g., by giving an example of classification tasks.
- I think it's better to add what kind of (neural) vocoder is used in the main body (not in the appendix) to asses the sound quality for their experiments.



**Experience Assessment:**

I have published one or two papers in this area.

**Review Assessment: Checking Correctness Of Derivations And Theory:**

I assessed the sensibility of the derivations and theory.

**Review Assessment: Checking Correctness Of Experiments:**

I assessed the sensibility of the experiments.

**Review Assessment: Thoroughness In Paper Reading:**

I read the paper at least twice and used my best judgement in assessing the paper.

---

> ### Author Response · Authors · 2019-11-14
> **Summary of Response to Reviewer #1**
>
> We thank the reviewer for their feedback.
> We agree with the reviewer that reproducibility is of vital importance. To help address the reviewers concerns we have:
>
> 1) Added experiments on an open data-set (libriTTS) to the paper.
> 2) Added much more detail about our wave-RRN vocoder
> 3) Added a table summarising all hyper-parameters to aid reproduction.
> 4) Added much more detail to the description of the CHBG text encoder
>
> We would also like to point out that very similar models have been successfully reproduced in the past. See here: https://github.com/NVIDIA/tacotron2 or here: https://github.com/Rayhane-mamah/Tacotron-2, Our model requires only a very modest increase in compute resources over these methods.
>
> We hope that our changes have helped to assure the reviewer of the reproducibility of our work and that they may revise their score because of this.

---

> > ### Author Response · Authors · 2019-11-14
> > **Detailed Response to Reviewer # 1 -- Part 1 of 2**
> >
> > >> My concern for this paper is a lack of reproducibility. The paper uses the in-house data to perform their experiments and the code does not seem to be publically available. Also, the paper misses several detailed information (e.g., detailed configurations of the Wave RNN vocoder, what kind of neural network toolkits and libraries). The high computational cost ("distributed across 32 Google Cloud TPU chips") would also make the reproducibility difficult.
> >
> > We agree that reproducibility is of vital importance to good research and to address your concern, have added further results using the publicly available Libritts dataset [1] on the continuous attributes such as F0 variation and speaking rate, with no tuning (exact hyperparams used for our internal dataset), and demonstrate the reproducibility of the method. These results are currently included in the appendix. But, we are happy to move them (along with MOS evaluations) to the main paper for the camera ready version, and add samples to our demo page.
> >
> > [1] Zen, H., Dang, V., Clark, R., Zhang, Y., Weiss, R. J., Jia, Y., ... & Wu, Y. (2019). LibriTTS: A Corpus Derived from LibriSpeech for Text-to-Speech. arXiv preprint arXiv:1904.02882.
> >
> > We have also added detail on the neural vocoder used and made all hyper-parameters more explicit.
> >
> > The only neural network library used was tensorflow 1 and we have added a mention of this to the paper, though we would argue that the choice of library used shouldn’t pose a serious challenge to reproducibility.
> >
> > On the point of requiring a large amount of compute resource, we train on 32 google cloud TPU chips primarily to accelerate iteration and experimentation, as this allows us to train in hours. It is perfectly possible to train a similar model on a single GPU within a few days and indeed there are numerous open source examples of very similar models e.g https://github.com/NVIDIA/tacotron2, https://github.com/Rayhane-mamah/Tacotron-2  that are able to do so. Our method involves a very modest increase in computation over existing methods that have already been reproduced in open source.
> >
> > >> I also would like to see whether this method can have some experimental comparisons with (Hsu et al., 2018) with their postprocessing to show the distinction in terms of the performance in addition to the functional difference
> >
> > This is an excellent question as the two works are very similar. One of the primary benefits of our method is that it is not necessary to perform post-processing in the manner of Hsu et al. In their work Hsu et al use fully unsupervised latent variables and so have to investigate the latent space manually in order to determine what the latent variables control. In our case the supervision determines the form of the latent variable so this is not necessary. Hsu et el, perform three forms of post-processing: 1) they look at intra-mean distances between latent modes, 2) they look at the assignment of labels between modes and 3) they search for dimensions of the latent that result in the greatest variation of the data. As we don’t have a latent mixture, there are no means for us to analyse and we already know the dimensions of variation for our continuous latents.
> >
> > Comments:
> >
> > - Figure 1: it's better to have an explanation of "CBHG". People outside the end-to-end TTS community cannot understand it.
> > >> We have added a description of the CBHG block to the appendix and referenced it from the paper.
> >
> > >> Can you also control the noise level as shown in (Hsu et al., 2018) but more explicitly within this framework? Controlling the noise level is quite important for end-to-end TTS, and I think this method can fit this direction because we can easily obtain the noise attribute (supervision) by data simulation or annotate the noise.
> >
> > This is a great suggestion and something that would be very natural to try with our method, either with a discrete clean/noisy or noise type categorical variable, or a continuous signal-to-noise ratio (SNR) level, as supervision. We have not explicitly run this experiment but believe it would make for interesting future work, and encouraged by the results of controlling both discrete and latent factors in this paper, we believe it is likely to work.

---

> > > ### Author Response · Authors · 2019-11-14
> > > **Detailed Response to Reviewer # 1 -- part 2 of 2**
> > >
> > > >>equation (6), classification loss: I think this part requires more clarifications in this timing, e.g., by giving an example of classification tasks.
> > > Thanks for your suggestion, We have amended the paper as follows (change in bold):
> > >
> > > “ As q(z_s|x, y, φ) is trained to approximate p(z_s|x, y, θ) we can expect it to become a reasonable classifier/regressor for the semi-supervised latent attribute as the model improves. For example when z_s represents an affect label, p(z_s|x, y, \theta) is the posterior probability of the model, over affect given  text and speech. By taking the most likely posterior class, this distribution can be used as an affect classifier. However, this variational distribution is only trained on unsupervised training points and so does not benefit directly from the supervised data. To overcome this problem we follow Kingma & Welling (2013) and add a classification loss to our objective.  ”
> > >
> > > >>I think it's better to add what kind of (neural) vocoder is used in the main body (not in the appendix) to asses the sound quality for their experiments.
> > >
> > > We have moved the description of the neural vocoder into the main body and expanded the detail in the appendix..

---

### Official Review · AnonReviewer3 · 2019-10-23
**Official Blind Review #3**

**Rating:** 6

**Review:**

The authors propose to do neural text to speech, conditioned on attributes such as valence and arousal and speech rate. A seq-to-seq network is trained using stochastic gradient variational Bayes.
The idea is interesting and new.

The method section could be made clearer by giving first some intuition, explaining the formulas in the prose and introducing the terms used.

Regarding the crowd sourced MOS: how were the ratings obtained? how many subjects were used for the rating? How were they selected? Was each rater presented with samples from each method? Or was each method assessed by different groups of raters?

The experimental setting is a little weak, relying mostly on the Mean Opinion Score. It would be useful to include more evalution, for instance:
* run a speech recognition method on the generated speech and measure the error rate

* examples could be included in the supplementary  (e.g. spectrograms)

* For the evaluation of emotional speech to be meaningful, the proposed classifier should be tested, and more detailed given (hyper-parameters, training setting, validation/test splits?, etc). In particular, it would be useful to compare the proposed method for affect classification to an existing (state-of-the-art) methodology. A state-of-the-art method should be ideally be directly used to classify the generated sequences into emotional classes.

The authors collected data in studio conditions, as such it is hard to compare. Will the data be released? How much hdata was collected?
It seems that what the authors effectively do is condition on discrete emotion classes, not valence and arousal, which are continuous measures of affect.

**Experience Assessment:**

I do not know much about this area.

**Review Assessment: Checking Correctness Of Derivations And Theory:**

I assessed the sensibility of the derivations and theory.

**Review Assessment: Checking Correctness Of Experiments:**

I assessed the sensibility of the experiments.

**Review Assessment: Thoroughness In Paper Reading:**

I read the paper at least twice and used my best judgement in assessing the paper.

---

> ### Author Response · Authors · 2019-11-14
> **Summary of Response to Reviewer #3**
>
> We thank the reviewer for their time and comments.
>
> For reasons outlined in detail below we do not fully agree with the statement that our initial evaluation was weak or relied primarily on MOS. However to help further improve our evaluation, we have extended the results by adding:
>
> 1) Automatic speech recognition word accuracy on our generations.
> 2) subjective human evaluation of affect controllability (see table 1 in the paper)
> 3) sample spectrograms in addition to the audio examples we previously provided.
> 4) results on an open data set (libriTTS)
>
> We hope this addresses the reviewer’s concern about our evaluations and that the reviewer may reconsider their score because of these improvements.

---

> > ### Author Response · Authors · 2019-11-14
> > **Detailed Response to Reviewer # 3  -- Part 1 of 2**
> >
> > [Summary of Paper]
> > The description of our main method is partially correct but we would like to highlight that our key contribution is to show that we can use probabilistic latent variables with a very *small* amount of supervision to control attributes of speech. This has not previously been possible.
> >
> > We demonstrate that we need as little as 30 minutes of supervision to control discrete latent factors such as arousal and valence categories, and as little as 3 minutes of supervision to control continuous factors such as speaking rate and pitch variation.
> >
> > >>The method section could be made clearer by giving first some intuition, explaining the formulas in the prose and introducing the terms used.
> >
> > Thanks, we have edited the paper for clarity.
> >
> > >>Regarding the crowd sourced MOS: how were the ratings obtained? how many subjects were used for the rating? How were they selected? Was each rater presented with samples from each method? Or was >>each method assessed by different groups of raters?
> >
> > Thanks for bringing this to our attention. We have now added much more detail to our description of MOS evaluation to the appendix.
> >
> > >>The experimental setting is a little weak, relying mostly on the Mean Opinion Score. It would be useful to include more evaluation, for instance:
> >
> > >>* run a speech recognition method on the generated speech and measure the error rate
> >
> > Please see point 3 below.
> > >>* examples could be included in the supplementary  (e.g. spectrograms)
> > Please see point 4 below.
> >
> > Overall we reject the assertion that we rely primarily on Mean Opinion Score for the following reasons:
> >
> > 1) The objective controllability metrics in Figure 4 are the primary metrics that we use to demonstrate the efficacy of our proposed method. The purpose of including MOS (which is considered a gold standard metric of quality) is simply to show that controllability does not come at the expense of speech synthesis quality. (We have now also added subjective metrics of control that also demonstrate our ability to control affect; please see Table 1 in the paper).
> >
> > 2) In addition to MOS, which is a subjective quality metric, we provide  Mel-Cepstral-Distortion-Dynamic-Time-Warping  (MCD-DTW) [1] scores. MCD-DTW is an objective measure of overall quality.
> >
> > 3)For most state-of-the-art TTS systems, the word error rate of automatic speech recognition systems are likely equivalently low and so this is not a particularly useful metric. This is why we did not include it. The frontier of TTS research is generally no longer focussed on word accuracy and indeed our focus is on controllability. We have nonetheless added both word and character error rates to our paper (please see Tables 2 and 3), where it can be seen that the error is indeed close to 0.
> >
> > 4) Whilst we did not have spectrograms in the paper we provided extensive audio samples on our demo page tts-demos.github.io. The samples were not cherry picked and even show transfer of control to speakers for whom we had no labelling. We have now added spectrograms to the appendix that clearly visualize some aspects of controllability. However, we would emphasise that speech examples are best listened to rather than viewed as images on paper.
> >
> >
> > >>* For the evaluation of emotional speech to be meaningful, the proposed classifier should be tested, and  more detailed given (hyper-parameters, training setting, validation/test splits?, etc). In particular, it
> >  would be useful to compare the proposed method for affect classification to an existing (state-of-the-art)  methodology. A state-of-the-art method should be ideally be directly used to classify the generated >>sequences into emotional classes.
> >
> > The purpose of evaluating synthesized speech with a classifier trained on ground truth, was to objectively show that we are able to synthesise speech in each of our emotional classes. That a relatively simple classifier is able to recognise emotions from our synthesised speech is surely sufficient to demonstrate control. In other words, we do not believe a SOTA affect classifier would add to our evaluation.
> >
> > The classification accuracy coupled with the audio samples we provide should allow readers to assess quite clearly how much control is achievable using our method. In order to further demonstrate controllability we have also added more subjective metrics of control to the paper in which we ask human raters to distinguish emotion of a synthesized utterance (please see Table 1).
> >
> > We acknowledge that more detail of the classification set-up including details of validation/test splits would aid reproducibility and have added these details in the appendix.

---

> > > ### Author Response · Authors · 2019-11-14
> > > **Detailed Response to Reviewer #3 - part 2 of 2**
> > >
> > > >>The authors collected data in studio conditions, as such it is hard to compare. Will the data be released? >>How much data was collected?
> > >
> > > The labelled data-set is proprietary and unfortunately can not be released. As stated at the top of section 3 (Data): “The training set consists of 72,405 utterances with durations of at most 5 seconds (45 hours). The validation and test sets each contain 745 utterances or roughly 30 minutes of data.”
> > >
> > > The completely unlabelled data that was used in the transfer experiments is from the 2013 Blizzard challenge and is publicly available.
> > >
> > > We have now included results on the public libriTTS dataset [1], with no tuning (exact hyperparams used for our internal dataset), and demonstrate the reproducibility of the method. These results are currently included in the appendix. But, we are happy to move to the main paper along with MOS evaluations on this data for the camera ready version, and to add samples to our demo page.
> > >
> > > [1] Zen, H., Dang, V., Clark, R., Zhang, Y., Weiss, R. J., Jia, Y., ... & Wu, Y. (2019). LibriTTS: A Corpus Derived from LibriSpeech for Text-to-Speech. arXiv preprint arXiv:1904.02882.
> > >
> > > >>It seems that what the authors effectively do is condition on discrete emotion classes, not valence and >>arousal, which are continuous measures of affect.
> > >
> > > The reviewer is correct that arousal and valence are generally continuous but were modelled as discrete in our paper. This was an explicit choice, since we had a small number of arousal-valence combinations (6 combinations). However, we do also demonstrate control using continuous latent variables for speaking rate and F0 variation. There is no reason in principle not to model arousal/valence as continuous and if one had genuine continuous variation in the training data and also a small amount of continuous labels, this would be sensible. Our method can be applied equally well to both continuous and discrete data.

---

### Comment · Area_Chair1 · 2019-11-15
**Reviewers, any comments on the author responses?**

Dear Reviewers, thanks for your thoughtful input on this submission!  The authors have now responded to your comments.  Please be sure to go through their replies and revisions, and let them know if you have additional feedback or questions.  Thanks!

---

### Decision · Program_Chairs · 2019-12-19

**Decision:**

Accept (Poster)

**Comment:**

The authors propose to enforce interpretability and controllability on latent variables, like affect and speaking rate, in a speech synthesis model by training in a semi-supervised way, with a small amount of labeled data with the variables of interest labeled.   The idea is sensible and the results are very encouraging, and the authors have addressed the initial concerns brought up by the reviewers.